# A Study of Shoot Growth, Leaf Photosynthesis, and Nutrients in 'Lingfengjing' Litchi Grafted onto Seedlings of Different Cultivars

**Yan Fan** [1,2], **Zhiyuan Li** [1], **Binxia Xie** [1], **Xiaowen Liang** [1,3] and **Xuming Huang** [1,*]

1   College of Horticulture, South China Agricultural University, Guangzhou 510642, China; fanyan198402@163.com (Y.F.); lizhiyuan1019@163.com (Z.L.); xiebinxia2022@163.com (B.X.); L1019420334@163.com (X.L.)
2   Dongguan Agricultural Technology Extension Center, Dongguan 523083, China
3   Guangxi Academy of Specialty Crops, Guilin 541004, China
*   Correspondence: huangxm@scau.edu.cn

**Abstract:** Litchi (*Litchi chinensis* Sonn.) is one of the important fruit crops in southern China. 'Lingfengnuo' (LFN) is a new late-maturing litchi variety which has gained great popularity among consumers and growers due to its high quality and reliable productivity. However, there has been limited research on the graft compatibility of LFN on different rootstocks, which is important for selecting the optimal rootstocks for propagation, and thus application of this new cultivar. In this study, LFN scions were grafted onto one-year-old seedlings from 13 cultivars including 'Heiye' (HY), 'Shakeng' (SK), 'Hehuadahongli' (HHDHL), 'Maguili' (MGL), 'Xiaojinzhong' (XJZ), 'Huaizhi' (HZ), 'Chenzi' (CZ), 'Shangshuhuai' (SSH), 'Baitangying' (BTY), 'Shuangjianyuhebao' (SJYHB), 'Jingganghongnuo' (JGHN), 'Baila' (BL), and 'Shuidong' (SD). Graft success, morphology of the graft joint, shoot growth, leaf photosynthetic rate, and nutrients were compared. Graft success was highest for XJZ, HZ, BL and JGHN. Tree vigor, reflected by flush growth, was highest for HZ, followed by JGHN and BTY, but weakest for HY, SD, SJYHB, and XJZ. The stem significantly swelled above the graft joint in trees with rootstocks of HY, SD, and SJYHB seedlings; however, this did not occur in XJZ. Leaf photosynthesis displayed a similar pattern to tree vigor, and was highest for HZ and lowest for HY, SD, and XJZ. These low vigor trees produced flowers the next year after grafting. The result suggested that HZ, JGHN, and BTY seedlings as rootstock are highly compatible with LFN, while those of HY, SD, and SJYHB are poorly compatible. Rootstocks did not significantly influence the leaf N, P, and Fe, but showed significant differences in other minerals. The leaf Ca, Mg, Zn and B were lowest in the trees graft onto HZ seedlings. Based on these results, HZ seedlings are recommended as the best rootstock for the propagation of LFN; however, a supply of Ca, Mg, B and Zn fertilizers a especially needed for LFN trees on HZ seedling rootstock.

**Keywords:** Lingfengnuo; litchi; rootstock; tree vigor; photosynthesis; minerals

## 1. Introduction

Litchi (*Litchi chinensis* Sonn.) is a Sapindaceae plant native to southern China and is a one of the major fruit crops in China, India and Vietnam [1]. Despite more than 200 litchi varieties recorded in China [2], commercial production in the country depends upon a dozen cultivars. In most countries outside Asia, 'Mauritius' and 'Tai So' are the dominate cultivars [3]. Even in China, the large-seeded cultivars, 'Huaizhi' (HZ), and 'Heiye' (HY), were extensively planted during the period of rapid litchi development from the early 1980s to the end of the 1990s due to their high and reliable productivity. Currently, the two cultivars account for nearly half of the total production [4]. It is a common problem in the litchi industry across the world that narrow cultivar diversity causes a concentrated harvesting season or seasonal overproduction, creating a great

burden in the marketing of this highly perishable fruit and price crushes [4]. The traditional small-seeded high-quality cultivars such as 'Nuomici' (NMC) and 'Guiwei' (GW) are erratic in flowering and bearing due to the high chilling requirement for flowering, thus requiring more intensive management. NMC is also particularly susceptible to excessive fruit drop and cracking. Therefore, the development of these high-quality cultivars is limited. The development of small-seeded, high-quality and reliably productive cultivars are highly desirable. 'Lingfengnuo' (LFN) was a chance selection from a seedling generated from open-pollinated HZ, likely with a pollinizer of NMC. It was certificated in 2010 by the Crop Cultivar Certification Committee of Guangdong Province. The fruit is highly chicken-tongued with thick, sweet, slightly fragrant flesh, brightly red with an appearance similar to NMC. LFN is a reliable productive cultivar and has a longer harvest period [5], higher photosynthetic capacity [6] (Fan et al., 2011), higher resistance to fruit cracking, and better postharvest performance [7] (Fan et al., 2014) than NMC. Due to these outstanding traits, the cultivar has quickly gained popularity among consumers and growers. LFN is thus nicknamed as "the second generation of NMC".

There is a high demand for new cultivars. Currently, LFN has been widely applied in litchi producing regions in China. Extension of this new cultivar depends on top grafting as well as the delivery of nursery stocks. Our field observation showed that LFN grows vigorously when top grafted onto adult trees of HZ, NMC, GW, 'Baila' (BL) and 'Baitangying' (BTY), but showed poor compatibility on 'Feizixiao' (FZX), HY, and 'Shuangjianyuhebao' (SJYHB) [8]. Nursery stocks of litchi are propagated with either air layering or grafting, but in China grafting is more commonly used [9,10]. Graft compatibility is a key factor for determining graft success. A number of studies concerning graft compatibility between litchi cultivars have been reported [8,11–17]. Results obtained by Chen et al., (2018) proved that grafting compatibility is closely related to the genetic relation between scion and rootstock cultivars [18]. Based on the available reports and field observation, graft compatibility among litchi cultivars is roughly divided into poor and good scion-rootstock combinations [10]. Poor graft compatibility leads to weak tree vigor and poor photosynthetic function [11,13,17]. The limited study on the rootstock effect in LFN showed that it has poor graft compatibility with HY, SJYHB, and FZX [8]. However, for propagation of litchi with grafting, juvenile seedling trees are used as rootstock. There have been no reports about the performances of LFN grafted onto seedlings of different cultivars. In this study, LFN scions were grafted onto seedlings of 13 cultivars, and the tree growth, morphology of the graft joint, leaf photosynthetic performance, and mineral nutrients were examined in order to discover the optimal rootstock for the propagation of this new cultivar. Efficient propagation of LFN is important for the application of this new cultivar and thus important for the diversification of commercial cultivars.

## 2. Materials and Methods

### 2.1. Materials

The experiment was carried out in the demonstrating orchard of the Dongguan Agricultural Technology Extension Center. During the harvesting season of 2014, seeds of 13 litchi (*Litchi chinensis* Sonn.) cultivars including HY, 'Shakeng' (SK), 'Hehuadahongli' (HHDHL), 'Maguili' (MGL), 'Xiaojinzhong' (XJZ), HZ, 'Chenzi' (CZ), 'Shangshuhuai' (SSH), BTY, SJYHB, JGHN, BL, and 'Shuidong' (SD) were collected from different litchi production regions. The seeds were sowed in wetted sand for germination at ambient temperature. When 4 leaves fully expanded, at least 10 seedlings from each cultivar with uniform height were selected and transplanted into a prepared bed at a distance of 30 cm × 30 cm and frequently irrigated to keep the soil moist and fertilized with 2–5 g of 15-15-15 compound fertilizer per seedling monthly. In March 2016, when the stem thickness at 30 cm above ground was thicker than 0.5 cm, 1-year-old budwoods were collected from the outer canopy of an 8 year old LFN tree in the same demonstrating orchard. Scions containing 3 nodes were grafted onto the seedlings at the trunk, 30 cm above the ground, using the whip grafting method. Due to the differential growth status of the rootstock seedlings, some

seedlings were too weak and not suitable for graft. The actual number of grafted seedlings is shown in Table 1.

**Table 1.** Numbers of grafted 'Lingfengnuo' trees, survived trees, and graft success rates on seedling rootstocks of different cultivars.

| Seedling Cultivar | Number of Grafted Trees | Number of Survived Trees | Grafting Success Rate (%) |
|---|---|---|---|
| HY | 12 | 5 | 41.7 |
| BL | 32 | 22 | 68.8 |
| SD | 12 | 5 | 41.7 |
| BTY | 11 | 6 | 54.5 |
| HZ | 21 | 18 | 85.7 |
| JGHN | 11 | 7 | 63.6 |
| SJYHB | 5 | 2 | 40 |
| CZ | 5 | 2 | 40 |
| MGL | 9 | 4 | 44.4 |
| SSH | 11 | 5 | 45.5 |
| XJZ | 8 | 8 | 100 |
| HHDHL | 15 | 4 | 26.7 |
| SK | 6 | 3 | 50 |

*2.2. Grafting Success and Tree Growth*

Surviving trees and dead trees were counted to get the survival rates of the grafted trees in the winter of 2016 (3 December). The tree vigor was reflected by tree height (measured from graft joint to the highest shoot terminal) and latest flush length, which was recorded with a tape ruler. The stem thicknesses immediately below and above the graft joint were measured with a vernier caliper. Trees were separately photographed. The flowering situation in each tree was observed on 10 March, when the total shoot terminals and flowering terminals in the trees were recorded.

*2.3. Leaf Greenness and Photosynthetic Rate*

Two to five of the most vigorous trees were selected based on the number of survived grafted trees in each cultivar for leaf SPAD and photosynthetic analysis on 3 and 4 December 2016. Three mature leaves from the latest flush of each selected tree were tagged for measuring SPAD with a SPAD-502 chlorophyll meter and then for the net photosynthetic rate with a TPS-II photosynthesis system under an artificial light intensity of 800 $\mu$mol m$^{-2}$ s$^{-1}$ and ambient carbon dioxide and temperature conditions (~23 °C). Measurements were taken from 8:30 to 11:00 a.m.

*2.4. Leaf Mineral Analysis*

After the photosynthesis parameters were collected, the tagged leaves were collected and brought back to the lab, washed with deionized pure water, and oven dried at 65 °C for 72 h. The dried samples were grinded into powder for analysis of minerals including nitrogen (N), potassium (K), phosphorous (P), calcium (Ca), magnesium (Mg), iron (Fe), copper (Cu), zinc (Zn), manganese (Mn), and boron (B). Sample preparation and analysis were carried out according to [19]. Half of a gram of the powder from each sample was digested in 5 mL of concentrated sulfuric acid at 420 °C. N concentration was determined with a Kjeltedc AUTO-2300 automatic nitrogen analyzer (Foss Tecator AB, Hoganas, Sweden). P was analyzed with the Mo-Sb-ascorbic acid colorimetric method, and K was analyzed using the flame photometry method. The other elements, Ca, Mg, Fe, Mn, Zn, Cu and B, were analyzed using a Z-5000 atomic spectrophotometer.

*2.5. Statistics*

The above measurements were carried out with at least two tree-based biological replicates. For the flush growth and flowering investigations, data from all of the survived

grafted trees were collected (*n* = number of survived trees). For measuring the photosynthesis, leaf greenness, and minerals, if a cultivar had more than 5 survived trees, the samples were taken from 5 trees of the cultivar (*n* = 5); if a cultivar had fewer than 5 survived trees, then the samples were taken from all of the survived trees (*n* = number of survived trees). Correlation and linear regression analyses and one-way ANOVA plus Duncan's multiple range tests were performed using SPSS 19 (Chicago, IL, USA).

## 3. Results and Analysis

### 3.1. Survival Rate of Grafted Trees on Different Rootstocks

Table 1 shows that among the 13 cultivars, HY, SD, CZ, SJYHB, HHDHL, and MGL had a graft success rate lower than 50%. The highest survival rates were found in XJZ, HZ, BL, and JGHN: 100%, 85.7%, 68.8%, and 63.6%, respectively.

### 3.2. Tree Growth

Tree height, latest flush length, stem thickness above and below graft joint, and rate of flowering terminals data are listed in Table 2 and the tree morphology is shown in Figure 1. Apparently, LFN grafted onto HZ seedlings had the strongest tree vigor, with the longest flush length and largest tree height among all rootstocks. Beyond HZ, flush length was highest for JGHN, BL, and CZ, and lowest for XJZ, SD. and SK. The tree height was highest for BTY, JGHN, BL, and SD but lowest in SK, SJYHB, HHDHL, and HY. Although the order of flush length and tree height among the rootstock cultivars had some differences, flush length had a strong positive linear correlation with tree height (Figure 2).

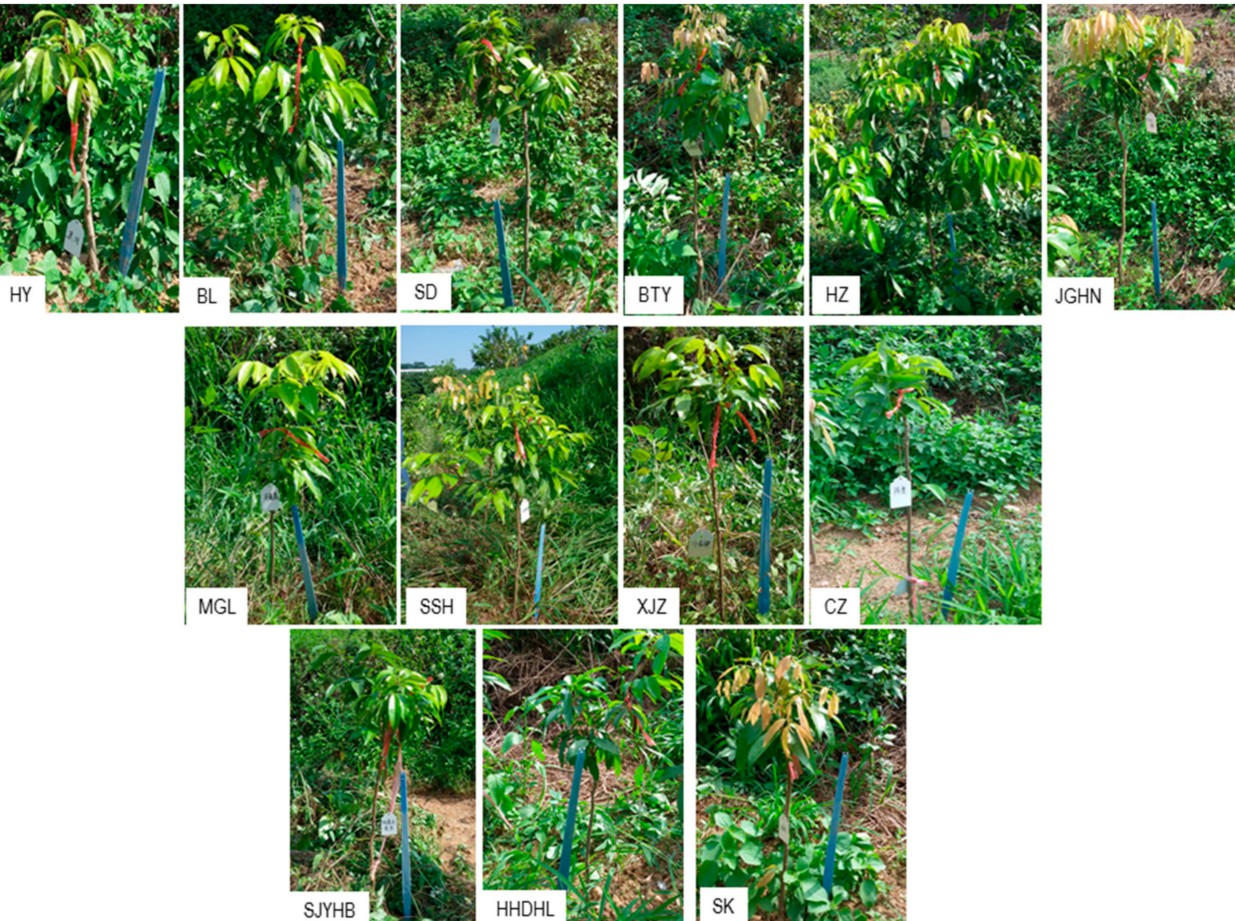

**Figure 1.** Morphology of 'Lingfengnuo' grafted onto seedlings of different cultivars.

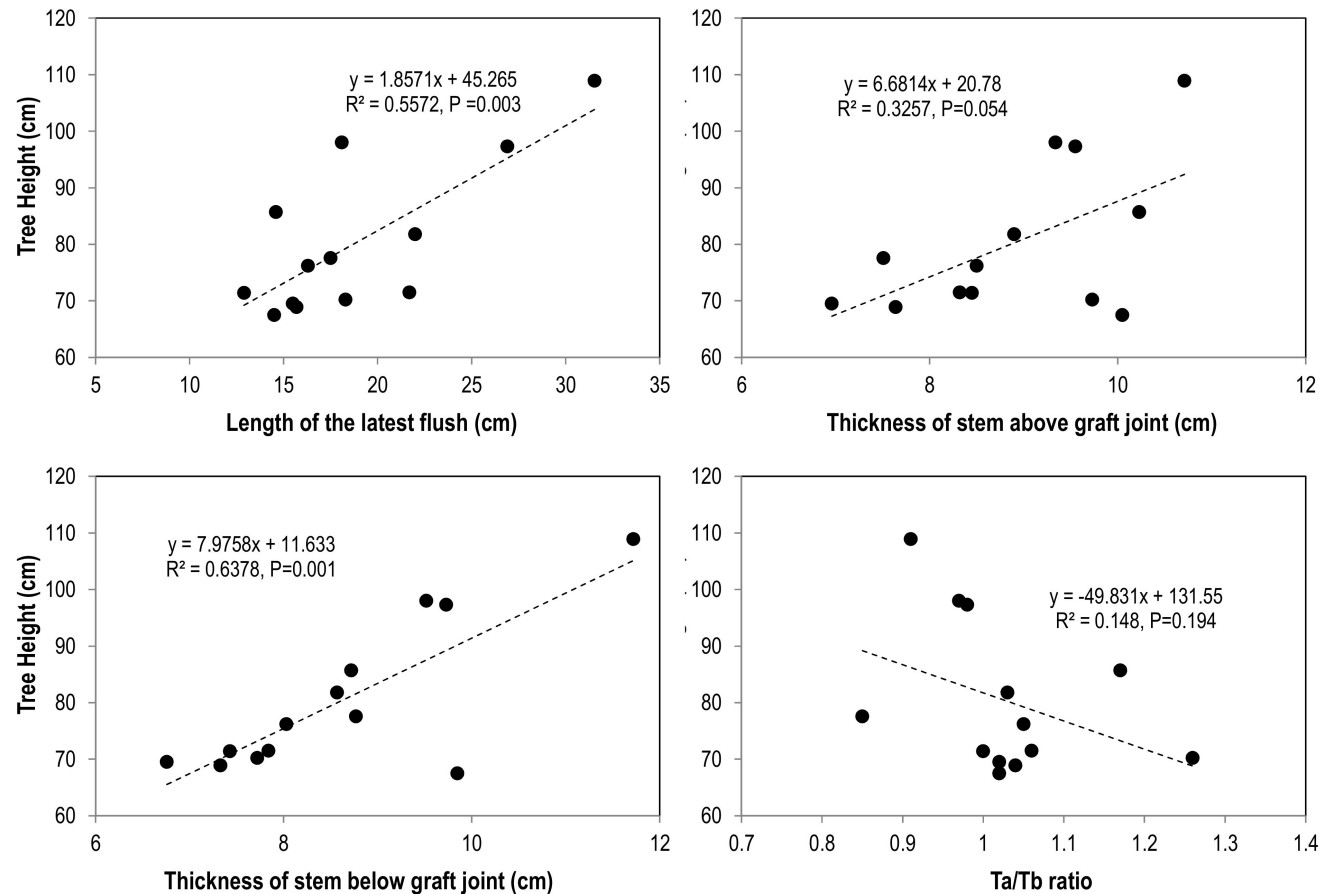

**Figure 2.** Linear correlation analyses between tree height and length of the latest flush, stem thickness above (Ta) and below (Tb) the graft joint, and the ratio of Ta/Tb.

**Table 2.** Flush growth and flowering of 'Lingfengnuo' grafted onto a different rootstock.

| Rootstock | Latest Shoot Length (cm) | Tree Height (cm) | Thickness above Graft Joint (Ta) (cm) | Thickness below Graft Joint (Tb) (cm) | Ta/Tb | Flowering Rate (%) |
|---|---|---|---|---|---|---|
| HY | 18.3 ± 4.4ab | 70.2 ± 7.9cd | 9.73 ± 1.26ab | 7.72 ± 0.99bc | 1.26 * | 34.4 |
| BL | 22.0 ± 1.9ab | 81.8 ± 5.4bc | 8.90 ± 0.79b | 8.57 ± 0.53bc | 1.03 | 15.4 |
| SD | 14.6 ± 3.4bc | 85.7 ± 5.6bc | 10.23 ± 1.18a | 8.72 ± 1.06bc | 1.17 * | 0 |
| BTY | 18.1 ± 1.9ab | 98.0 ± 6.0ab | 9.34 ± 1.07ab | 9.52 ± 0.84ab | 0.97 | 7.0 |
| HZ | 31.4 ± 4.9a | 108.9 ± 8.5a | 10.71 ± 1.24a | 11.74 ± 1.15a | 0.91 * | 0 |
| JGHN | 26.9 ± 5.4ab | 97.3 ± 9.7ab | 9.55 ± 1.00ab | 9.73 ± 0.93ab | 0.98 | 0 |
| SJYHB | 16.3 ± 3.2ab | 76.2 ± 5.2cd | 8.50 ± 0.54b | 8.03 ± 0.30bc | 1.05 | 0 |
| MGL | 21.7 ± 8.9a | 71.5 ± 10.9cd | 8.32 ± 1.41b | 7.84 ± 0.09bc | 1.06 * | 0 |
| SSH | 15.7 ± 3.2bc | 68.9 ± 5.0d | 7.64 ± 0.68bc | 7.33 ± 0.61bc | 1.04 | 0 |
| XJZ | 12.9 ± 2.9c | 71.4 ± 2.8cd | 8.45 ± 0.58b | 8.43 ± 0.73bc | 1.00 | 25 |
| CZ | 17.5 ± 7.2ab | 77.5 ± 7.2cd | 7.51 ± 1.88bc | 8.77 ± 0.70bc | 0.85 | 0 |
| HHDHL | 15.5 ± 5.2bc | 69.5 ± 0.5d | 6.96 ± 1.31c | 6.76 ± 1.08c | 1.02 | 0 |
| SK | 14.5 ± 3.1bc | 67.5 ± 7.5d | 10.05 ± 2.61a | 9.85 ± 0.21ab | 1.02 | 0 |

Different letters after mean ± standard error indicate a significant difference of $p < 0.05$, Duncan's multiple range test. * indicates that this value is significantly different from 1.0 with a $p < 0.05$, two-tailed $t$-test.

Trees with HZ seedling rootstock also had the highest stem thickness both below and above the graft joint. Stem thickness above the graft joint (Ta) was in the following order: HZ > SD > SK > HY > JGHN > BTY > BL > SJYHB > XJZ > MGL > SSH > CZ > HHDHL (Table 2). Ta had a weak positive correlation with tree height (Figure 2).

Stem thickness below the graft joint (Tb) followed the following pattern: HZ > SK > JGHN > BTY > CZ > SD > BL > XJZ > SJYHB > MGL > CZ > SSH > HHDHL (Table 2). Ta had a strong positive linear correlation with tree height (Figure 2).

The ratio of Ta/Tb was highest in HY (1.26), followed by SD (1.17), MGL (1.06), and SJYHB (1.05). Trees with these rootstocks had a swelled graft joint, which is a sign of poor graft compatibility. The lowest value was found in CZ (0.85) and HZ (0.91). The remaining rootstock treatments were close to 1.0. Ta/Tb displayed an insignificant negative linear correlation with tree height (Figure 2).

After the chilling winter in 2016, some of the grafted trees produced panicles. Shown in Table 2, 34.4%, 25%, 15.4%, and 7% of the shoot terminals produced flowers in LNF trees with rootstocks of HY, XJZ, BL, and BTY seedlings, respectively. Trees with the other rootstocks produced no flowering.

### 3.3. Leaf Greenness and Photosynthetic Capacity

Leaf greenness reflected by SPAD value was highest in trees with rootstocks of HHDHL, CZ, SK, and HZ seedlings, and lowest for XJZ, SSH, and HY. The other rootstock groups showed no significant differences (Table 3). The net photosynthetic rate (Pn) was generally low during the winter season (Table 3) and differed significantly among rootstocks. Trees with HZ, MGL, and SK rootstocks, and JGHN seedlings, had the highest photosynthetic rate, while those grafted onto seedlings of HY, SD and XJZ had the lowest photosynthetic capacity. Pn had a significant positive linear correlation with flush length and stem thickness below the graft joint (Tb), but a significant negative linear correlation with Ta/Tb and was not significantly correlated with SPAD and stem thickness above the graft joint (Ta) (Figure 3).

**Table 3.** Leaf greenness and net photosynthetic rate among trees with different rootstocks.

| Rooststock Cultivar | Leaf Greenness (SPAD) | Net Photosynthetic Rate ($\mu mol \ m^{-2} \ s^{-1}$) |
|---|---|---|
| HY | 29.6 ± 1.76cd | 1.24 ± 0.26c |
| BL | 35.3 ± 2.57bc | 2.17 ± 0.22bc |
| SD | 32.2 ± 2.26bc | 1.75 ± 0.41c |
| BTY | 33.2 ± 1.74bc | 4.14 ± 1.01b |
| HZ | 37.3 ± 1.32ab | 7.18 ± 0.86a |
| JGHN | 33.4 ± 1.44bc | 4.80 ± 1.01ab |
| SJYHB | 31.0 ± 1.34bc | 3.24 ± 0.60bc |
| MGL | 32.3 ± 3.05bc | 5.66 ± 1.35ab |
| SSH | 25.9 ± 2.05d | 4.38 ± 0.65b |
| XJZ | 25.7 ± 1.27d | 2.18 ± 0.60bc |
| CZ | 43.1 ± 1.16a | 4.11 ± 1.07b |
| HHDHL | 42.1 ± 3.18a | 3.12 ± 0.70bc |
| SK | 39.5 ± 2.02ab | 5.37 ± 0.63ab |

Different letters after mean ± standard error indicate a significant difference at *p* < 0.05, Duncan's multiple range test.

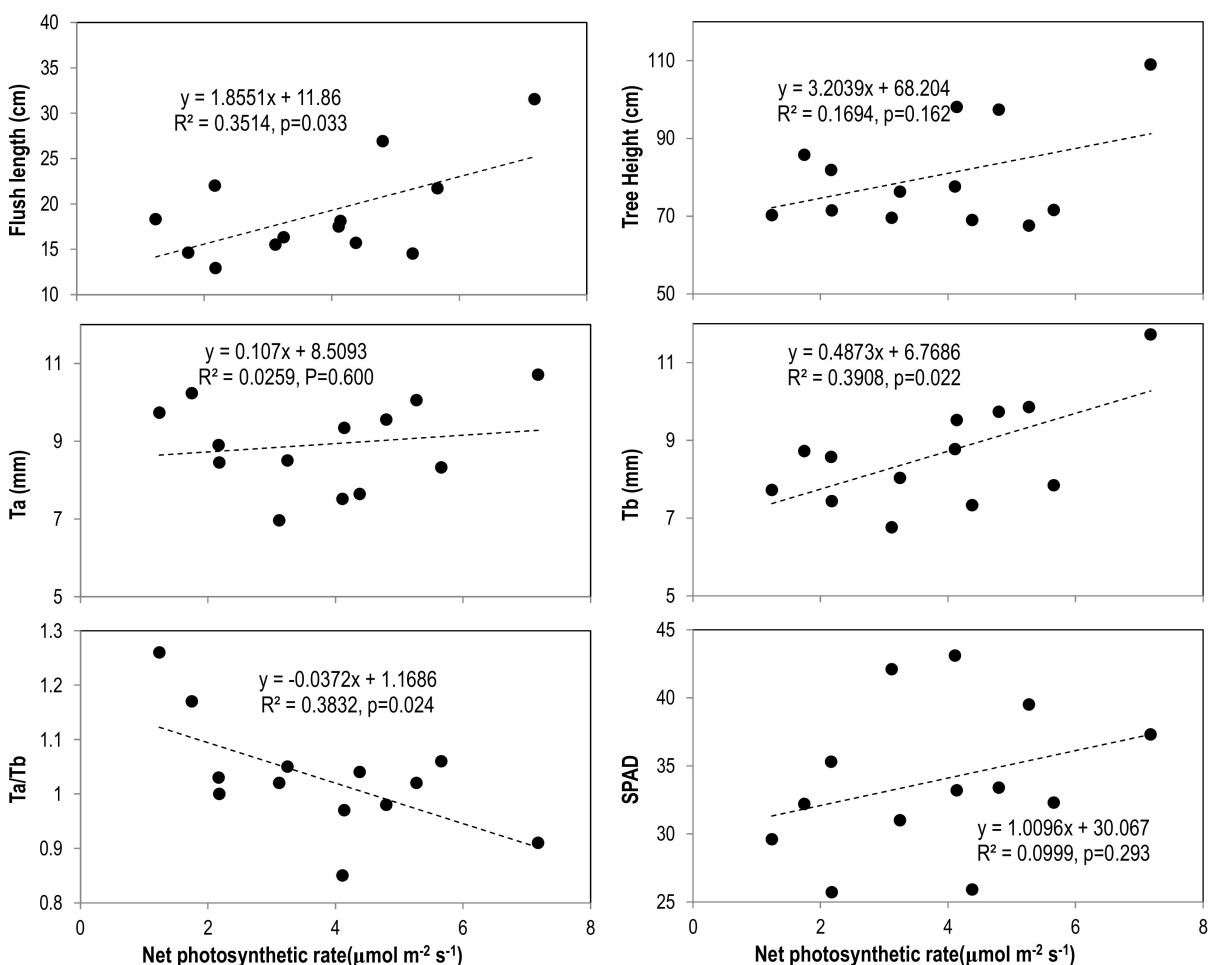

**Figure 3.** Linear correlation analysis of net photosynthetic rate with flush length, tree height, stem thickness above (Ta) and below (Tb) graft joint, and leaf SPAD value.

### 3.4. Leaf Mineral Nutrients and Their Correlations with Tree Vigor and Photosynthesis

The macronutrients of N and P were similar among all rootstocks varieties (Table 4). HY, BL, and SJYHB had the highest K concentration, which was significantly higher than the lowest value found in SK. The remaining rootstock groups had no significant differences. Ca was highest for SK and XJZ, and lowest for HZ and SJYHB. The former was significantly higher than the latter two rootstock varieties. Mg was highest for XJZ (2.58 g/kg), which was significantly higher than most of the other rootstocks, and lowest in YZ and CZ (<1.0 g/kg).

Among the micronutrients (Table 5), Fe was similar across all rootstocks. Cu was the highest in SJYHB and the lowest in SD, but both had no significant differences with the other rootstocks. Zn, Mn, and B differed significantly among rootstocks. XJZ had the highest Zn, followed by SK and HHDHL. HZ and HY had the lowest Zn concentration. Mn was highest for SK, followed by HHDHL, MGL and CZ. BL and HY had the lowest Mn concentration. B was highest for XJZ followed by CZ and SSH. HZ had the lowest B concentration. Except for Ca and Fe, which had a significant negative correlation with tree height, none of the minerals had a significant correlation with tree height, flush length, or stem thickness (Table 6). Only P displayed a significant positive correlation with Pn. Leaf SPAD value was significant positively correlated to N but negatively to Mg. N was negatively correlated with Mg and Fe, P was negatively correlated with Cu, K displayed a strong negative correlation with Mn and Ca, and Ca was positively correlated with Zn, Fe and Mn (Table 6).

**Table 4.** Macronutrients in the leaves of 'Lingfengnuo' grafted onto different rootstocks.

| Rootstock Cultivar | N | P | K | Ca | Mg |
|---|---|---|---|---|---|
| | (g/kg) | (g/kg) | (g/kg) | (g/kg) | (g/kg) |
| HY | 14.53 ± 0.76a | 1.12 ± 0.06a | 13.16 ± 0.43a | 2.86 ± 0.14ab | 1.72 ± 0.14ab |
| BL | 14.20 ± 0.85a | 1.14 ± 0.03a | 12.62 ± 2.27a | 3.14 ± 0.34ab | 1.54 ± 0.15b |
| SD | 13.75 ± 2.27a | 0.97 ± 0.06a | 11.43 ± 0.71ab | 3.36 ± 0.43ab | 1.77 ± 0.31ab |
| BTY | 15.03 ± 1.25a | 1.22 ± 0.12a | 10.48 ± 1.13ab | 3.31 ± 0.74ab | 1.75 ± 0.48ab |
| HZ | 15.12 ± 0.51a | 1.38 ± 0.15a | 8.98 ± 0.60ab | 2.07 ± 0.15b | 0.91 ± 0.13b |
| JGHN | 14.87 ± 0.86a | 1.45 ± 0.16a | 11.6 ± 0.79a | 2.96 ± 0.37ab | 1.71 ± 0.08ab |
| SJYHB | 16.20 ± 4.92a | 1.54 ± 0.61a | 12.56 ± 0.51a | 2.23 ± 0.26b | 1.38 ± 0.12b |
| MGL | 13.44 ± 2.09a | 1.49 ± 0.16a | 9.89 ± 0.66ab | 3.77 ± 0.12ab | 1.19 ± 0.26b |
| SSH | 11.86 ± 0.68a | 1.17 ± 0.04a | 9.34 ± 0.53ab | 3.74 ± 0.46ab | 1.77 ± 0.10ab |
| XJZ | 12.74 ± 2.00a | 1.13 ± 0.33a | 9.85 ± 2.17ab | 4.12 ± 0.79ab | 2.58 ± 0.53a |
| CZ | 15.45 ± 2.04a | 1.25 ± 0.12a | 7.77 ± 3.51ab | 3.55 ± 1.49ab | 0.99 ± 0.04b |
| HHDHL | 16.19 ± 0.34a | 1.44 ± 0.27a | 8.18 ± 2.10ab | 3.85 ± 0.51ab | 1.01 ± 0.01b |
| SK | 14.74 ± 2.76a | 1.41 ± 0.16a | 6.07 ± 1.21b | 4.72 ± 0.10a | 1.36 ± 0.18b |

Different letters after mean ± standard error indicate a significant difference at $p < 0.05$, Duncan's multiple range tests.

**Table 5.** Micronutrients in leaves of 'Lingfengnuo' grafted onto different rootstocks.

| Rootstock Cultivar | Cu | Zn | Fe | Mn | B |
|---|---|---|---|---|---|
| | (mg/kg) | (mg/kg) | (mg/kg) | (mg/kg) | (mg/kg) |
| HY | 9.46 ± 0.80ab | 22.21 ± 0.47cde | 82.17 ± 24.03a | 37.46 ± 3.09de | 25.43 ± 3.18bc |
| BL | 9.92 ± 0.53ab | 26.79 ± 2.84bcde | 69.64 ± 8.73a | 32.03 ± 7.21e | 19.83 ± 3.32bc |
| SD | 8.61 ± 0.24b | 24.95 ± 1.20bcde | 72.29 ± 16.24a | 43.29 ± 2.36cde | 14.70 ± 3.59c |
| BTY | 10.58 ± 1.36ab | 21.15 ± 0.83de | 59.99 ± 9.10a | 72.52 ± 16.78bcde | 15.38 ± 1.50c |
| HZ | 11.28 ± 0.96ab | 20.56 ± 1.20e | 59.22 ± 3.93a | 71.95 ± 9.11bcde | 14.14 ± 3.59c |
| JGHN | 15.22 ± 1.84ab | 27.06 ± 0.44bcde | 57.39 ± 3.79a | 76.39 ± 8.91bcde | 24.32 ± 3.43bc |
| SJYHB | 17.49 ± 7.45a | 29.61 ± 5.10bcd | 52.42 ± 7.41a | 46.02 ± 10.30cde | 16.89 ± 8.91bc |
| MGL | 15.72 ± 1.62ab | 27.71 ± 2.49bcde | 62.81 ± 9.13a | 110.33 ± 8.25b | 28.51 ± 1.32abc |
| SSH | 9.50 ± 0.92ab | 22.66 ± 1.35cde | 86.45 ± 5.01a | 89.81 ± 5.4b | 30.52 ± 2.45ab |
| XJZ | 15.07 ± 3.15ab | 37.65 ± 3.13a | 87.73 ± 16.35a | 86.80 ± 17.38bc | 42.37 ± 6.03a |
| CZ | 13.45 ± 6.46ab | 24.17 ± 6.65cde | 59.15 ± 3.93a | 104.13 ± 35.65b | 31.25 ± 9.06ab |
| HHDHL | 13.93 ± 2.22ab | 30.37 ± 3.83abc | 69.94 ± 2.73a | 120.48 ± 26.97b | 18.02 ± 1.48bc |
| SK | 10.06 ± 3.72ab | 33.31 ± 0.88ab | 91.12 ± 28.52a | 169.49 ± 20.42a | 23.53 ± 1.95bc |

Different letters after mean ± standard error indicate a significant difference at $p < 0.05$, Duncan's multiple range tests.

**Table 6.** Correlations between measured parameters.

| | FL | TH | Ta | Tb | Ta/Tb | SPAD | Pn | N | P | K | Ca | Mg | Cu | Zn | Fe | Mn | B |
|---|---|---|---|---|---|---|---|---|---|---|---|---|---|---|---|---|---|
| FL | 1 | | | | | | | | | | | | | | | | |
| TH | 0.746 ** | 1 | | | | | | | | | | | | | | | |
| Ta | 0.423 | 0.571 * | 1 | | | | | | | | | | | | | | |
| Tb | 0.679 * | 0.799 ** | 0.751 ** | 1 | | | | | | | | | | | | | |
| Ta/Tb | −0.315 | −0.385 | 0.237 | −0.427 | 1 | | | | | | | | | | | | |
| SPAD | 0.187 | 0.133 | −0.089 | 0.330 | −0.486 | 1 | | | | | | | | | | | |
| Pn | 0.593 * | 0.412 | 0.161 | 0.625 * | −0.619 * | 0.316 | 1 | | | | | | | | | | |
| N | 0.196 | 0.259 | 0.020 | 0.261 | −0.246 | 0.700 ** | 0.095 | 1 | | | | | | | | | |
| P | 0.330 | 0.028 | −0.174 | 0.150 | −0.364 | 0.353 | 0.622 * | 0.525 | 1 | | | | | | | | |
| K | 0.164 | 0.182 | 0.206 | −0.176 | 0.556 * | −0.542 | −0.552 | −0.036 | −0.265 | 1 | | | | | | | |
| Ca | −0.625 * | −0.623 * | −0.328 | −0.365 | −0.021 | 0.092 | −0.069 | −0.384 | −0.127 | −0.621 * | 1 | | | | | | |
| Mg | −0.401 | −0.073 | 0.031 | −0.238 | 0.124 | −0.704 ** | −0.394 | −0.608 * | −0.465 | 0.228 | 0.303 | 1 | | | | | |
| Cu | 0.070 | −0.087 | −0.426 | −0.217 | −0.329 | 0.011 | 0.181 | 0.315 | 0.697 ** | 0.027 | −0.128 | 0.002 | 1 | | | | |
| Zn | −0.481 | −0.524 | −0.247 | −0.362 | −0.024 | −0.056 | −0.201 | −0.060 | 0.218 | −0.246 | 0.557 * | 0.434 | 0.479 | 1 | | | |
| Fe | −0.545 | −0.603 * | 0.017 | −0.330 | 0.400 | −0.322 | −0.309 | −0.604 * | −0.461 | −0.272 | 0.671 * | 0.404 | −0.488 | 0.376 | 1 | | |
| Mn | −0.222 | −0.357 | −0.251 | 0.025 | −0.392 | 0.443 | 0.514 | 0.027 | 0.447 | −0.913 ** | 0.735 ** | −0.172 | 0.140 | 0.423 | 0.315 | 1 | |
| B | −0.333 | −0.528 | −0.424 | −0.438 | −0.102 | −0.366 | −0.129 | −0.574 * | −0.158 | −0.199 | 0.517 | 0.463 | 0.266 | 0.473 | 0.484 | 0.286 | 1 |

* and ** indicate correlations are significant at $p < 0.05$ and $p < 0.01$, respectively. FL, TH, Ta, and Tb represent length of the latest flush, tree height, stem thickness above and below graft joint, respectively.

## 4. Discussion

### 4.1. 'Lingfengnuo' Shows Differential Graft Compatibility with Seedlings of Different Cultivars

Rootstocks exert a strong effect on tree performance in multiple aspects, including tree vigor, flowering and fruiting behavior, quality formation, and resistance to biotic and abiotic stresses [20]. In fruit production, dwarfing rootstocks that lower tree vigor are highly desirable for permanently controlling tree size for efficient tree management and harvesting. In litchi, there have been no commercial dwarfing rootstocks. However, a number of studies have shown that rootstocks have a significant effect on shoot growth and bearing performance [9,13–17]. The effect of rootstock on tree vigor in litchi is very much related to scion-rootstock compatibility, and poor compatibility results in a low tree vigor and the yellowing of leaves [10,18,21]. Understanding scion-rootstock compatibility is crucial for the application of new cultivars, especially in regions where new cultivars are applied chiefly via top grafting or graft propagation, such as China. Incompatible grafting is reflected by the poor quality or failure of vascular connection between the rootstock and the scion, leading to low efficiency in material exchange between the root and top [17,22–24]. The supply of root-born resources, such as water and minerals, to the shoots may be limited due to incompatibility, leading to poor shoot growth and poor leaf function, e.g., photosynthesis. Therefore, shoot growth and leaf photosynthetic capacity are useful indicators to evaluate graft compatibility.

The new litchi cultivar, LFN, is a reliable crop with superior quality [5]. The limited study on top grafting showed that it had relatively poor grafting compatibility with HY, SJYHB, and FZX and highly was compatible to HZ, NMC, GW, BTY, and BL [8]. In this study, we further examined the performances of LFN grafted onto the seedlings of 13 cultivars. Graft success rate (survival rate) ranged from 26.7% to 100%. The highest survival rates were found in XJZ, HZ, BL, and JGHN and the lowest were found in HHDHL, CZ, SJYHB, and HY. Although graft success is subject to influence by multiple factors such as climate, operator grafting skills, budwood status, etc. [21]. Scion-rootstock compatibility might play a crucial role in graft success in this experiment, since grafting was performed by the same person and under the same climate and tree management conditions. Chen et al., (2017) showed that poor scion-rootstock compatibility resulted in slower healing and higher risk of grafting failure [25]. Viewing from the perspective of graft success, seedlings of XJZ, HZ, BL, with JGHN as the rootstock had a relatively high compatibility with LFN, while those of HHDHL, HY, CZ, and SJYHB were of relatively low compatibility. Despite that, XJZ had the highest graft success (100%). The trees on XJZ seedlings had the lowest tree vigor, reflected by the smallest flush length and tree height. There was also no obvious stem swelling at the graft joint. This result suggests that XJZ seedlings as rootstock may have a dwarfing effect on LFN. This effect needs further observation. Trees on HY, SD, SJYHB, and MGL seedling rootstocks showed significant stem swelling above the graft joint, an indicator of poor compatibility. Their tree vigor was also generally lower than those on the other rootstocks. Seedlings of these cultivars as rootstock are thus of low compatibility with LFN, which agrees with the observations made in top-grafting trees [8]. LFN grafted onto HY and SD also had the lowest net photosynthetic rate. In contrast, tree vigor was highest for rootstocks of HZ, BTY, JGHN, and BL, and these scion-rootstock combinations showed no stem swelling at or above the graft joint. Net photosynthetic rate was also relatively high for HZ, BTY, and JGHN. These were highly compatible with LFN. Based on sequence-related amplified polymorphism (SRAP) analysis [26], LFN had a very close genetic relation with HZ and JGHN, but a distant one from HY, SD, and SJYHB. Therefore, scion-rootstock compatibility based on graft success, tree vigor, graft joint morphology, and photosynthesis supports the hypothesis that graft compatibility is closely related to the genetic relation between the scion and rootstock [18].

It is worth to note that LFN grafted onto CZ seedlings tended to produce a swelled rootstock stem with the smallest Ta/Tb ratio (0.85) among all cultivars. This is another phenotype of an incompatible graft. However, it did not cause a significant reduction of

tree vigor when compared with the other compatible rootstock cultivars besides HZ. The mechanism of rootstock stem swelling needs to be further explored.

It is also worth noting that some scion-rootstock combinations with low tree vigor, such as trees with HY, SD, and XJZ, produce flowers within one year after grafting, indicating that low tree vigor is in the favor of flower differentiation. Therefore, vegetative growth controlling measures need to be taken to enhance flowering in LFN on such invigorating rootstocks as HZ seedlings.

### 4.2. Rootstocks Exert Influence on Mineral Nutrients

Numerous studies have shown that rootstocks have a significant effect on mineral nutrient levels and composition [27–30]. The mineral difference caused by rootstocks might be attributed to two aspects. The first aspect is the graft compatibility effect: Scion-rootstock compatibility is associated with nutrient (e.g., $^{32}$P and $^{14}$C) exchange between the root and top [27,28]. As previously discussed, an incompatible graft results in a poor vascular connection between the root and top, which definitely affects the mineral supply from the root to the leaves. The results in our study showed that the rootstocks had a significant influence on all of the minerals tested, with the exceptions of N, P, and Fe (Table 4). We could not see any clear association between minerals and graft compatibility. The leaf mineral concentration may not directly reflect compatibility, as it is also affected by the volume of flush growth, i.e., the dilution effect due to growth. Tree vigor seemed to have no significant correlation to leaf minerals, except for Ca and Fe, which were negatively correlated to tree height. Yet, the leaves of the most vigorous LFN trees which were grafted onto HZ seedlings had the lowest Ca, Mg, Zn, and B levels, while the opposite was seen for the weakest trees grafted onto XJZ seedlings. The mechanism behind this pattern needs further clarification. A second aspect related to difference in minerals caused by rootstocks might be differential preference in mineral uptake among the roots of different rootstocks, which is worth further exploration. The results of our study suggested that the application of Ca, Mg, Zn, and B fertilizers is particularly necessary for LFN trees grafted onto HZ seedlings.

### 5. Conclusions

LFN has high graft compatibility with seedlings of HZ, JGHN and BTY but a poor compatibility with seedlings of HY, SD and SJYHB. Among the rootstocks tested, HZ seedlings are the best for propagating LFN as they generated the highest tree vigor with highest photosynthetic capacity. However, supply of Ca, Mg, B and Zn fertilizer is especially need for LFN trees grafted onto HZ seedlings.

**Author Contributions:** Conceptualization, methodology and validation, Y.F. and X.H.; formal analysis and investigation, Y.F., Z.L., B.X., X.L. and X.H.; Resources, Y.F.; data curation, Y.F., Z.L., B.X. and X.L.; writing—original draft preparation, Y.F. and Z.L.; writing—review and editing, X.H.; supervision, project administration and funding acquisition, X.H. All authors have read and agreed to the published version of the manuscript.

**Funding:** This study was funded by National Natural Science Foundation (31772249) and National Litchi and Longan Research System (CARS-33-11).

**Institutional Review Board Statement:** Not applicable.

**Informed Consent Statement:** Not applicable.

**Data Availability Statement:** Not applicable.

**Acknowledgments:** The authors acknowledge Yanjie Fa and Hongyan Zhang (South China Agricultural University) for assisting with gas exchange measurements.

**Conflicts of Interest:** The authors declare no conflict of interest.

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
