# Peer review of "A Study of Shoot Growth, Leaf Photosynthesis, and Nutrients in ‘Lingfengjing’ Litchi Grafted onto Seedlings of Different Cultivars"

_horticulturae, doi:10.3390/horticulturae8040282_

Round 1

Reviewer 1 Report

The paper describe the grafting potential/compatibility on different rootstock of Litchi (Litchi chinensis Sonn.), which is one of the important fruit crops. It is believed  that grafting compatibility is closely related to genetic relation between scion and rootstock cultivars and it can be categorized as poor and good scion-rootstock combinations. Poor graft compatibility leads to weak tree vigor and poor photosynthetic function.  Thus, the authors well justified the studied topic. The paper described and compared for the graft success, morphology of graft joint, shoot growth, leaf photosynthetic rate and nutrients accumulation. The authors demonstrated that rootstocks did not influence leaf N, P and Fe. Though MS can be accepted as is, I have following suggestions to further improve the MS.
1. Discussion: The authors described the effect of grafting compatibility on photosynthetic rate. However, the authors did not correlate with possible mechanism. 
2. Similarly, how change in rootstock affect nutrient uptake and accumulation of some macronutrients but it did not affect N, P and micronutrient Fe. 

Author Response

Many thanks for reviewers comments and suggestions. As suggested we have added lines to address relation between photosynthesis and compatibility. Also, minerals in relation to rootstock is addressed. We added one more citation revealing poor vasicular connection at the graft junction which affects substance exchanges and thus leaf functions.

Reviewer 2 Report

Section 2.1 How many seedlings and how many budwoods were used?

Section 2.2. What mean photoed.?

Section 2.3 (Photosynthesis): indicate the time and temperature and the number of leaves used to determine CO2 assimilation.

Section 2.3 (Leaf minerals): change 2.3 to 2.4. Indicate the references for the methods used.

Figure 1. The authors must use the same acronyms used throughout the text to facilitate reading in the images. For example, what is HZ in Figure 1?

In (Ta/Tb) the correlation is not significant; it is incorrect to mention "had a weak positive correlation."

In 3.3. change "was not insignificantly" to "was not significantly".

In Figure 3, it is incorrect to use Net photosynthetic rate, the correct is CO2 assimilation rate.

Author Response

Many thanks to the reviewers for constructive suggestions.  We have revised our manuscript largely based on the suggestios. The followings are our responses to reviewer's concern and suggestions:

(1) We displayed numbers of seedlings planted for grafting in Table 1, which differed among cultivars, but at least 10 seedlings were transplated for each cultivars. However, tree growth status differed greatly. Some died after transplanting, some were too weak for grafting. Therefore, many cultivars had less than 10 trees grafted.  Budwood number was the same as grafted tree as grafted made on trunk of the seedlings. 

(2) photographed.

(3) Revised as suggested.

(4) Revised as suggested

(5) The statment has been revised into "Ta/Tb displayed an insignificant negative linear correlation with tree height".

(6) Revised as suggested.

(7) Our instrument measures net photosynthetic rate, which exludes release of CO2 by respiration, instead of CO2 assimilation rate. In all our previous publications we used net photosynthetic rate. Therefore, we insist using net photosynthetic rate.

Reviewer 3 Report

Revision

Abstract and introduction well set out the context, the importance of the species and the relevance of the study.

Materials and methods: “The leaves were washed with “clean water”... It is distiller water? Otherwise, the water could contribute minerals to the sample.

Statistical Analysis: “with at least three biological replicates” this seems very limiting and means that in some cases it was only 2 replicates to photosynthesis and growth parameters.

Results and analysis:

Why in some cases graft 32 trees on a rootstock and in other cases only 3 trees? 100% of grafting success rate with only 3 trees grafted it is unreliable…

“Ta/Tb displayed a weak negative linear correlation with tree height (Figure 2)”: With a R2 = 0.148 and P=0.194, there is no linear correlation.

Author Response

Many thanks to the reviewers for critical comments and constructive suggestions, to which we have the following response:

  1. Abstract and Introduction: We have specified that our study is important for selecting optimal rootstocks for propagation and thus application of this new cultivar. 
  2. We mean "Clean water"  deionized pure water. We have revised accordingly.
  3. Statistics. You are right. Suvival trees in some rootstocks were less than 3.  We intially wanted to have at least 5 tree-based replicates with grafted tree number no smaller than 5 each cultivar.  But in reality, some cultivars had less than 3 survised grafted trees.  In addition, we checked our orignal record and found that HHDHL's data was mixed with that of another cultivar Jidi, which had only 1 tree survived out of 6 grafted and had been excluded in this paper. Actually, HHDHL had 4 out of 15 grafted trees survived. Another mistake we identified was that XJZ had 8 trees instead of 3 trees grafted and all trees survived. We have made correction for the two cultivars in Table 1. Description of statistics has been revised detailing biological replciates for different independents.  It has revised as "The above measurements were carried out with at least 2  tree-based biological replicates. For flush growth and flowering investigation, data from all survived grafted trees were collected (n=number of survived trees). For measuring photosynthesis, leaf greenness and minerals, if a cultivar had more than 5 survived trees, the samples were taken from 5 most tallest trees of the cultivar (n=5); if a cultivar had fewer 5 survived trees, then samples were taken from all the survived trees (n=number of survived trees). Correlation and linear regression analyses and one-way ANOVA plus Duncan’s multiple range tests, which were performed using SPSS 19."
  4.  The great difference in grafted tree number was because of different seed number collected. The number of seed collected in some non- commercialized minor cultivars was small (some as few as 20, such as XJZ, Shaken) , while others such as BL, HY and HZ are cultivated widely and it was easy to collected their seeds.  Seed of different cultivars are collected from different places and showed great differenct in viability. Number of relatively uniform seedlings differed. We were able to select at least 10 trees each cultivar for transplanting. Yet seedling vigor differed. A few of them died after transplanting and a number too weak to carry out graft. All these caused difference in numbers of grafted trees among cultivars.  As mentioned above, the number of 3 grafted trees for XJZ was mistaken and it was 8 grafted trees.  
  5. Ta/Tb displayed an insignificant negative linear correlation with tree height.